# Conformity Consumption Behavior and FoMO

**Inwon Kang, Haixin Cui and Jeyoung Son ***

Department of International Business & Trade, Kyung Hee University, 26 Kyungheedae-ro, Dongdaemun-gu,
Seoul 02447, Korea
* Correspondence: sonjeyoung@khu.ac.kr

**Abstract:** Culturally associated brands that reflect the cultural backgrounds of other countries have been extensively studied, but research on the conformity consumption of culturally associated brands in specific countries has rarely been conducted. This study focuses on the fear of missing out (FoMO) phenomenon as a tool for explaining consumers' conformity consumption and examines what causes it. Although the FoMO concept is primarily used in the online field, it is considered to be a very suitable tool for explaining offline consumption behavior as well. This study will be useful for establishing long-term and sustainable strategies by firms through matured discussions on conformity consumption.

**Keywords:** fear of missing out; conformity consumption; culturally associated brand; evaluation process

## 1. Introduction

In recent years, it appears that non-normal collective consumption phenomena focused on specific brands occur in a particular country or generation, in which people follow or imitate the consumption behavior of others around them. For instance, in order to wear the same gear as one's companions, a frenzied buying wave of Canada Goose, a popular clothing brand from Canada, has emerged in the Korean clothing market over the past few years [1]. Likewise, the American outdoor brand, The North Face, which is known as a "second school uniform" among Korean middle school and high school students, has become an essential item with mass consumption by teenagers [2]. The preference for these culturally associated brands reflects the cultural background of a particular country [3] and depicts excessive following-up consumption by consumers in certain Korean groups. The following-up behavior toward a certain brand under the influence of others is considered as conformity consumption in the academic field [4,5]. It deeply reflects the values and actions of others and shows the effect of stimulation from the consensus of consumers.

Then why are consumers obsessed with the culturally associated brands above? Previous studies suggest that the conformity tendency can not only be affected by relatively general factors such as price, quality, and uniqueness compared with competing brands [6,7], but also by the psychological motivation of individuals, such as susceptibility to the behavior, lifestyle, and influences of the people around them [8–10]. In addition, the strong a specific culture also leads to the preference for the culturally associated brand, as seen with the influence of Korean wave, Otaku culture in Japan, and American hippy culture [10,11].

However, it is hard to use these factors alone to explain the excessive conformity consumption of Canada Goose and The North Face among Korean consumers, because it tends to be an irrational consumption to some extent. First of all, the prices for these brands wares are as high as luxury brands and have created a large economic burden for some consumers, but the preference for them is still increasing. In addition, regardless of the price or quality and whether it is suitable or not, everyone

owns at least one. If someone does not have one, they can even be discriminated against by the people around them. As such, the frenzied consumption of these brands is not simply due to the preference for the brand itself or the culture of particular country. The abnormal consumption of these brands appears to be a following and imitating behavior under the influences of mainstream groups. It is more illustrative to explain this excessive conformity consumption in terms of the intense feeling of being excluded a consumer has when separated from the mainstream group in which they are involved. Therefore, there is a limitation in using previous studies merely to explain the excessive conformity consumption phenomena. It is necessary to identify the motives properly.

Some studies point out that individuals are often influenced by others' choices and tend to follow the common beliefs and lifestyle of mainstream groups [8,12]. The feelings arise from the desire to integrate into mainstream society and the fear of being separated. This can be explained through the concept of fear of missing out (FoMO), which describes a psychological state in which people worry about losing touch with some social events, experiences, and interactions around them [13,14]. Research related to FoMO has mainly described it as a kind of mental state and emotional change which could lead to excessive social media, smartphone use, and alcohol consumption [15–17]. This research has revealed that FoMO has both direct and indirect effects on people's attitudes and behaviors. In other words, because of FoMO, people show a strong tendency and willingness to change their behavior to follow and imitate the collective or the group, which reflects the desire for not separating from the mainstream and of being the same as others. As such, there is good reason to consider FoMO as a useful tool for explaining the excessive consumption of culturally associated brands that appears in certain countries.

Therefore, this study examines the FoMO phenomenon and aims at understanding the role of FoMO in affecting the evaluation of culturally associated brands that lead to conformity consumption. To do this, we first introduce the concept of FoMO to explain conformity consumption. Then, we apply the FoMO phenomenon to the evaluation process model for the culturally associated brand. To empirically test the evaluation model, we focus on the Korean market, which is extensively experiencing the conformity consumption phenomenon. We selected the clothing brands The North Face, Canada Goose, and Moncler as survey subjects, as these are representative of culturally associated brands and have been extremely popular among Korean consumers. Recognizing the effect of FoMO on conformity consumption through the evaluation process will improve the explanatory power of pointing to the excessive conformity consumption of culturally associated brands. This will be of practical significance for company managers in formulating brand marketing strategies for guiding consumption behavior.

## 2. Theoretical Background

### 2.1. Conformity Consumption of Culturally Associated Brands

As members of a society, individuals tend to be influenced by the opinions, attitudes, and beliefs of the group to which they belong, and they are likely to consciously or unconsciously consume according to group values and norms [18]. The consumption behavior of an individual is influenced by society in their decision-making process and this influencing of other people's decisions as one of the important factors in individual decision making is referred to as conformity consumption in academic research [4]. It has been seen as a strong reflection of the degree of sympathy from others and it is considered that beliefs, values, and behaviors of others are influential [19].

Previous studies on conformity consumption illustrated that, on the one hand, brand characteristics such as quality, price, and uniqueness, compared with competing brands, have an influence on conformity consumption [6,7]. On the other hand, personality traits and stimuli from outside also have a primary influence on conformity consumption. Liang and He [20] argue that consumers follow the consumption of others to pursue symbolic or hedonic values, explaining that the main reason for conformity consumption behavior is due to the psychological motivation of the individual. In addition, Kastanakis and Balabanis [8] found that the common beliefs and lifestyles of mainstream groups lead

to conformity consumption for consumer. Park and Feinberg [10] explained that the opinions of the people around them are the decisive variables that cause an individual's conformity consumption. Dong and Zhong [21] suggested that jointly engaging in synchronous activities can activate a general "copying others" mindset, resulting in increasing conformity tendencies.

As culturally associated brands become more common and familiar to consumers, in some countries, these become the leading brands in society for a certain generation. Culturally associated brand refers to the brand often used as symbolic resources for the construction and maintenance of identity [22]. These brands represent the central concept it is connected to and the special characteristics of the culture [23]. Brands such as Armani from Italy and Sony from Japan are well-known brands that have become symbols of the cultures or countries with which they are associated. In the past, culturally associated brands introduced products that emphasized the technical skills of certain countries, such as electronics, high-end luxury clothing, and automobiles. Today, however, culturally associated brands circulate around the world in almost every product category [3].

There is an especially excessive conformity consumption of certain brands. For example, Canada Goose, a Canadian clothing brand, has become a somewhat freakish trend among Korean young people. Despite the large economic burden on some families, with the price of one suit exceeding 900 dollars, purchases are still rushing headlong. This phenomenon appears to be the excessive following-up consumption of a particular brand, rather than just the temporary favoring of some brands that are popular with consumers. A large segment of consumers has a general tendency to select some particular culturally associated brands consistently, despite their higher prices and sometimes even lower quality [24]. As such, it is difficult to explain how the excessive conformity consumption of these culturally associated brands appeared in some countries by the use of factors related to brand characteristics and interpersonal influences as presented in previous studies. The preference for some certain culturally associated brands appears to be a behavior under social pressure or influenced by an intense desire to not be isolated and discriminated against by the mainstream group in which they are involved. We see that it has become necessary to identify the motives that could properly explain the excessive conformity consumption.

### 2.2. The Role of FoMO in Explaining Excessive Conformity Consumption Behavior

#### 2.2.1. FoMO Phenomenon

Previous Studies of FoMO

Existing studies dealing with FoMO have been published in various academic fields, including psychology, management information system (MIS), and medicine [13–15,25–27]. Specifically, studies on the FoMO phenomenon in the field of psychology mainly describe FoMO as a kind of severe symptom like obsession toward the mainstream [13,17]. In the MIS field, FoMO is related to an emotional change, which leads to excessive use of social media and smartphones [15,25,26]. In addition, FoMO has been held as an addictive psychological symptom in medical fields such as psychopathology and psychiatry [27]. Additionally, a small number of studies have paid attention to alcoholism or psychological obsessive symptoms in consumption behavior, suggesting that FoMO could prompt consumption by putting pressure on consumers' decisions that lead to compulsive or impulsive behavior [16,17].

The above studies commonly describe FoMO as the basis for strong obsessive symptoms for certain subjects. As a consequence of this, early FoMO studies mainly focused on the anxiety or concern of being separated from the mainstream and reported that this attitude caused addictive behavior [13]. However, many recent studies have shown that FoMO can not only be reflected in the psychological anxiety of being separated from the group, but also in the case of reflecting FoMO from the social relations side, as well as the strong desire of the individual who wants to belong to the mainstream [15,16,25,26]. For example, Abel et al. [15] argue that FoMO is divided into three dimensions, as follows: Sense of self, social interaction, and social anxiety. Additionally,

Lai et al. [25] considered need to belong and social pain as two aspects of FoMO. In summary, it has been discussed in terms of a desire aspect that commonly belongs mainly to the group and anxiety that one may be excluded from the group [13,26]. Therefore, FoMO is generally considered to be a common experience in the field of psychology, which describes people's fear of detachment and the desire to stay continually connected with what others are doing [14]. It reveals an individual's desire for knowing and understanding what is going on.

Dimensions of FoMO

According to existing studies [15,16,25,27], FoMO was both linked to general psychological needs (basic needs) and social needs of belonging. Belongingness needs exist inherently among human beings. The need to belong has multiple impacts on people's cognitions, emotions, and behaviors. Oberst et al. [28] proposed that the general fear of missing out on something seems to be more associated with a need to belong. The FoMO could influence one's motives for participating. Individuals with high levels of the need to belong pay more attention to their interpersonal interactions and social connections with others.

In addition, some studies have examined FoMO in relation to variables involving the underlying negative affectivity and emotional distress dimension of psychopathology that drives a lot of depressive and anxiety disorders [29]. FoMO has generally revealed small to moderate correlations between negative affecting and mood [13], depression severity [28,30,31], and impaired behavioral activation associated with depression [31]. These depressions and disorders are reflections of the anxiety of isolation. To sum up, FoMO is not only a material and spiritual need, but also an anxiety embodied in people's behavior, which reflects the fear of detachment [25]. On the basis of previous studies, FoMO can mainly be distinguished by two psychological traits. One is the desire for belonging, which refers to the person experiencing the fear of missing out as having a strong need for interpersonal attachments as a fundamental human motivation, such as the need to identify with their status and need for recognition from others [32]. The other one is the anxiety of isolation, described as the fear of being broken away and being isolated from the mainstream group [13,33].

As for the dimensions of the desire for belonging, Elliott [34] suggested that individuals' desire for belonging to a mainstream group is due to the need for expressing the symbolic meaning of personality inherent in basic needs, which could be recognized as the dimension of prestige sensitivity and social presence. For instance, according to Oberst et al. [28], they argued that the person who has a greater attention towards their image in the mind of others and shows a higher need of recognition from others can be led to belongingness in the mainstream. Furthermore, Lai et al. [25] proved that individuals with a high level of fear of missing out had greater attention and propensity to meet their social connection needs. The connectedness dimension mainly refers to interactions with individuals and social environments [14,28]. The consciousness of other people also reflects the need to be connected with the people around oneself [32]. For this reason, many previous studies have regarded the connectedness as one of the sub-factors of the desire for belonging, and those who perceive high connectedness feel that they have a strong aspiration for the mainstream group [14,15,25]. Meanwhile, since praise from others reflects an individual's willingness to gain commendation and favor from other people, the expectancy of gaining praise from others may reflect an individual's desire to be constantly involved in the process of following novelties [13,15]. People show a strong desire for the acceptance and recognition from others in society. As a result, this study considers that prestige sensitivity, connectedness, and praise from others are dimensions of FoMO, which reflects the psychological trait of the desire to belong.

In previous studies related to FoMO, the anxiety of isolation has been described as a kind of concern about being separated from the group [15,25]. Wegmann et al. [14] pointed out that a person's core characteristics, such as their psychopathological symptoms, depression, and anxiety lead to a high level of FoMO in avoiding separation. In addition, Elhai et al. [31] found that people can slip into depression and anxiety when they are alienated from the people around them. Abel et al. [15]

suggested that the fear of social exclusion and the fear of ostracism may motivate people to be consistent with groups, largely in an attempt to avoid either exclusion or ostracism. The fear of being alienated and being ignored may lead to more promotion-focused responses, including reengagement in social contact, thoughts about actions one should have taken, and increased feelings of anxiety, which could be seen as dimensions of FoMO. According to Beyens et al. [32], adolescents with a stronger need for popularity experienced more FoMO of social media usage. The dimension falling behind is a psychological state that leaves one afraid of lagging behind other people or out of the flow of time. It can be seen as a pursuit of popular trends which maintains great attention to new things that are different from other people or to following popular trends. Therefore, being alienated, being ignored, and falling behind could be seen as the main dimensions for the anxiety of isolation. The details of each dimension constituting FoMO are shown in Table 1 below.

**Table 1.** Dimensions of FoMO.

| Category | Dimensions | Elliott [23] | Przybylski et al. [13] | Abel et al. [15] | Beyens et al. [32] | Lai et al. [26] | Wegmann et al. [14] | Oberst et al. [28] | Elhai et al. [31] |
|---|---|---|---|---|---|---|---|---|---|
| Desire for belonging | Prestige sensitivity | √ | | | √ | √ | √ | √ | |
| | Connectedness | | √ | √ | | √ | √ | √ | √ |
| | Praise from others | | √ | √ | | | | | |
| | Consciousness of other people | | | | √ | | | | |
| | Social presence | √ | | | | | | | |
| Anxiety of isolation | Being alienated | | √ | √ | | | √ | √ | |
| | Being ignored | | √ | √ | | √ | | √ | |
| | Falling behind | √ | | | √ | | √ | | |
| | Disapproval from others | | √ | | | | | | |
| | Depression and concern | | | | | | | | √ |

### 2.2.2. FoMO and Conformity Consumption Behavior

Consumer conformity has been defined as compliance with group norms, susceptibility to group influence, and behavioral changes in consumption behavior due to a reference group [6]. People usually possess a willingness to comply with others and will easily change their decisions under the influence of other people. In this manner, consumers are likely to change their mind or build their opinion about a specific brand, which can increase their concerns for that brand. In a situation where the stimulus information needs to be reflected upon, consumers' emotional condition is also integrated into the processing of the request, which is highly motivated by the perceived influence of the people around them. In other words, consumers conform to the expectations of others regarding purchase decisions and learn about product acceptability by observing the purchasing behavior of others [35].

Studies of FoMO suggest that it is a social phenomenon that leaves people feeling alienated from their current experiences, especially the young [31]. It is easy to make consumers feel unintegrated when they break away from the group of members of society. In order to resolve this anxiety, consumers develop a stronger tendency to imitate and follow the behavior of others [33]. Due to the desire to enter into the mainstream, the conformity consumption of culturally symbolic brands like The North Face and Canada Goose in a particular country can be seen as consumption influenced by the social consumption trend. As a consequence of the fear of being isolated from the mainstream and the desire for security, consumers display an increasing need for conformations from their surroundings [36]. As a result, FoMO, which encourages excessive use of a particular subject, can exert pressure on consumers to make decisions. In other words, FoMO portrays the psychology that consumers do not want to be excluded from the surrounding people and, therefore, behavior change based on these kinds

of psychological cues might also be utilized in investigating consumption behavior. It is appropriate to regard FoMO as a kind of mental state of individuals that imitates or follows others' behaviors blindly, which could be a useful tool for explaining excessive conformity consumption behavior.

*2.3. Evaluation Process of Culturally Associated Brand Based on FoMO*

This study applies the phenomenon of FoMO as a tool to explain a consumer's conformity consumption of culturally associated brands. Theories dealing with consumer behavior suggest that consumers experience changes in attitudes according to their beliefs and values prior to actual behavior [37]. In other words, the consumer experiences beliefs and values according to experience accumulated or external stimuli, thus leading to emotions or perceptions of the internal state of the individual. As a representative example, according to the social facilitation theory, consumers' attitudes are activated by influences from outside people who are external beings, and the attitudes are activated by specific responses [37]. Many studies that have applied social facilitation theory have mainly focused on online consumption behaviors and have reported that people are influenced by the groups they follow as the main motive for activating their activities [37,38]. In other words, the psychological state of an individual who wants to follow mainstream groups or those around them can be a major cause of the individual's active attitude.

These individual attitudes are expressed as complex emotions perceived for a specific object and, through the coordination of emotions, the future behavior is determined. In this context, emotion refers to the physiological responses of an individual to a specific object. Existing studies regard positive and negative emotions as independent bipolar dimensions and explain that the opposite of positive emotion is not negative emotion. In other words, consumers can co-exist with positive and negative emotions in decision making and the final evaluation of the object is made by coordinating the two emotions [39–41].

The two kinds of psychological traits based on FoMO, 'desire for belonging' and 'anxiety of isolation' have been presented as variables in previous studies. Through the different kinds of emotions perceived by consumers, there are different processes for determining future actions. Positive emotions of consumers are mainly assurance, favorability, likability, and preference, while negative emotions are mainly explained by animosity, antipathy, and uncertainty, etc. [38–40]. Desire for belonging can be seen as a positive emotion that is understood as psychological sureness through consumption and anxiety of isolation is the negative emotion which could be understood as psychological apprehension.

This stability and concern are strengthened through the process of activation. Stability refers to the individual's response to the emotions or circumstances that the consumer feels in adapting to their daily lives and to a comfortable state that feels enough to satisfy an individual's desire [42]. In other words, the more consumers perceive that they belong to a mainstream group and that they are recognized by others, the more likely they are to strengthen their stability in the group. On the other hand, a concern means a state of overburdening physiological or psychological to damage [29]. Consumers are increasingly concerned by the fear of being isolated if they perceive that they are excluded from the mainstream group. This psychology can be activated by attitudes and behaviors emphasizing the need to follow the mainstream group, which ultimately raises interest in the culturally associated brand.

As a result, consumer interest in a specific brand increases or decreases depending on the level of perceived stability of the individual or an increase in concern [43]. Through the activating process, consumers' growing interest in brands could be a clue to determine the direction of future conformity consumption behavior [44,45]. The overall evaluation process is shown in Figure 1.

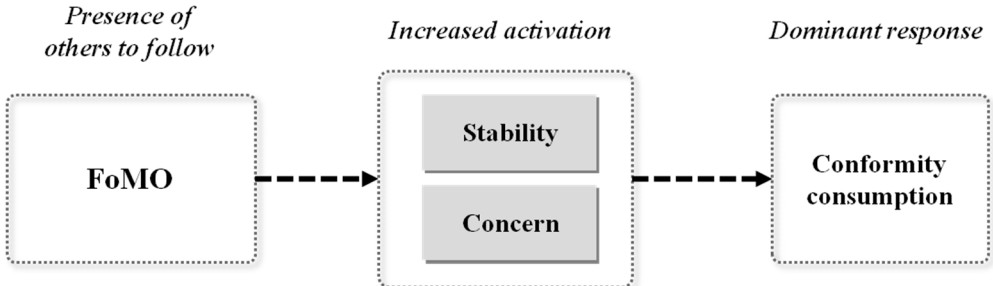

**Figure 1.** Conceptual framework of the evaluation process of the culturally associated brand.

## 3. Research Hypotheses

### 3.1. FoMO and Reinforcing Stability

Reinforcing stability refers to the state in which the propensity of consumption or service of a particular brand is steadily strengthened due to the consumer's desire for belonging. It means that a consumer who pays attention to their status or self-identity would have a quality of prestige sensitivity inside and would have a quick reflect to the products that could show position or status of them. Consumers who are highly prestige sensitive purchase an expensive brand not because of quality perceptions per se, but because of their perception that relevant others will perceive the purchase decision as reflective of their traits and social economic status [43]. Therefore, because of the strong needs of prestige sensitivity, consumers have a stability evaluation on a specific brand.

On the other hand, consumers may obtain the information through interactions with others or by directly observing social connections in their group. They may follow the purchases made by their social connections to keep up with what others have achieved through the product. Research has focused on how social connections and interactions can be exploited to promote consumers' attitudes and emotions, spending, and product adoption [46]. According to Aral and Walker [47], online social connections can promote the adoption of products and social interactions can stimulate spending on products. Ranaweera and Jayawardhena [46] suggested that interactions may lead to a change in consumer's attitudes.

In addition, studies have found that the praise and punishment of others greatly influences decisions or changes of an individual's behavior. Customers tend to try on new things, styles, and fantasize, wrapped in the anonymity of a self-service environment to fill their desire for belonging into the environment and desire to gain satisfaction from outside [48]. As a result, in the process of making decisions, the praise from others has a positive impact on the continued purchase of a particular brand and reinforces the stability of the purchase of the brand. Based on the discussion above, this study makes the following hypotheses:

**Hypotheses 1 (*H1*).** *Prestige sensitivity has a positive effect on reinforcing stability.*

**Hypotheses 2 (*H2*).** *Connectedness has a positive effect on reinforcing stability.*

**Hypotheses 3 (*H3*).** *Praise from others has a positive effect on reinforcing stability.*

### 3.2. FoMO and Increasing Concern

Fear of being alienated refers to a kind of uneasiness that consumers generate in order to cater to the consumption trends of the surrounding people in the process of making consumption decisions. The feeling of alienation, although emotionally discomfiting when it occurs, is actually a reflection of a unique aspect of human cognition with the use of representational abilities to enable a cognitive critique of our beliefs and our desires [49]. Therefore, the anxiety of being alienated can lead to the increasing concern of customers for culturally associated brands.

Fear of being ignored not only reflects the desire for not being neglected from close friends, but also from the social environment in which they are involved. It reveals the condition of one experience when they are neglected by others. When analyzing consumption experiences, one cannot ignore the social context within which experiences are shaped and lived [49]. Being ignored could lead individuals to seek to reestablish the contact and opportunities in order to get rid of this condition. As a consequence, they pay attention to the ways to increase concerns with surroundings and the change in the attitude toward buying choice, along with the mentation.

Diener [50] suggested that the wish not to fall behind is a greater spur to consumption than the wish to get ahead. In other words, sometimes people buy things not because they need them or to be leaders of other people, but rather in order to not fall behind others. The concern of falling behind appears in every aspect of life and society. In order to be constantly updated on the scene and to keep up with new things without falling behind others, new consumption patterns are acquired along with increasing concern on a new choice. Therefore, the following hypotheses are developed:

**Hypotheses 4 (*H4*).** *Being alienated has a positive effect on increasing concern.*

**Hypotheses 5 (*H5*).** *Being ignored has a positive effect on increasing concern.*

**Hypotheses 6 (*H6*).** *Falling behind has a positive effect on increasing concern.*

### 3.3. Consumer Emotional Change and Up-Surging Interest on Culturally Associated Brands

Up-surging interest on culturally associated brands reflects the change of evaluation and the intensive intention due to the reinforcing stability and increasing concerns about the culturally associated brands. In most situations, individuals are susceptible to the behavior of outsiders. Park and Feinberg [10] propose that attitudes are developed throughout the duration due to a learning process and are impacted by other people's influences, such as the social group in which the consumer is involved and the one to which individual aspires to belong, the received information, the experience, and the personality. The evaluation of people around a certain brand subconsciously has an effect on the predictable consumer behavior in more immediate future decisions.

In addition, people tend to have a growing interest in the brands accepted by the mainstream due to the willingness to belong to the majority and to be recognized by others. The desire for keeping stability in the situation and group increases the interest on the specific brands. On the other hand, concerns about the fear of being separated or isolated from the mainstream group could lead individuals to find the opportunities to avoid being far away from the condition and establish contact with the surroundings [51]. As a result, the negative psychological state of concern about being excluded from others would strengthen consumers' interests in specific brands. Therefore, this study attempts to make the following hypotheses:

**Hypotheses 7 (*H7*).** *Reinforcing stability has a positive effect on up-surging interest on culturally associated brand.*

**Hypotheses 8 (*H8*).** *Increasing concern has a positive effect on up-surging interest on culturally associated brand.*

### 3.4. Up-Surging Interest on Culturally Associated Brand and Conformity Consumption

Consumer conformity is defined as compliance with group norms, susceptibility to group influence, and behavioral changes in consumption behavior due to a reference group. Cheung and Prendergast [35] suggested that consumers wish to use products and brands which are popular amidst their peers. Consumers conform to the expectations of others regarding purchase decisions and learn about product "acceptability" by observing the purchasing behavior of others. There is a want for dissimilarity from others and yet a desire to be similar to others in one's social group. This blend in or stand out struggle exists because people mimic others in a social group in order to be accepted in that group themselves [52].

Generally, if a product provides better value than others, then consumers' purchase intention of this product will be high [43]. The reinforcing stability and the increasing concern, which can cause growing interest in the brand or products, are capable of leading to the consumption of conformity. Culturally associated brands are generally considered to be much more applied by consumers around the world and suffer a deficit compared to national brands with regard to the social symbolic or image benefits they offer. The increasing interests on the culturally associated brands may increase consumer's benefits and, as a result, have an obvious positive affect on the purchasing choice. Therefore, the following hypothesis is developed:

**Hypotheses 9 (*H9*).**　*Up-surging interest on culturally associated brand has a positive effect on conformity consumption.*

On the basis of the theoretical backgrounds described above, the model of this research is shown in Figure 2.

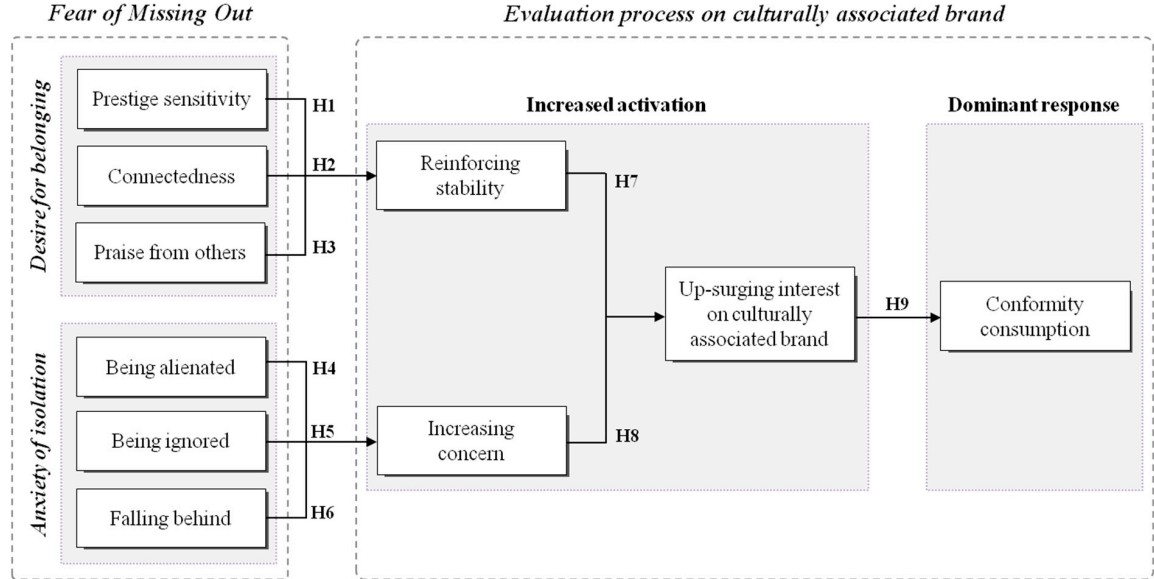

**Figure 2.** Research Model.

## 4. Methodology

### 4.1. Sampling

This study conducted a preliminary questionnaire before the survey. The preliminary questionnaire was distributed to 25 graduate students in Seoul, Korea. In this process, the wrong expression or difficult expression was finally checked and revised. The questionnaire survey was conducted for consumers living in Korea and quota sampling was used to collect a similar sample of gender and age. The subjects of the survey were consumers who have purchased representative clothing brands such as Moncler, Canada Goose, and The North Face. The reason for choosing clothing as the culturally associated brands to be investigated is because they are products that are exposed to the outside and easily reflect the trend of society [53–55]. In addition, Moncler, Canada Goose, and The North face were selected because they are meaningful as representative culturally associated brands that have been greatly popular in Korea in recent years [1,2].

The questionnaire was conducted as an offline survey. In the course of conducting the survey, respondents were briefed on the culturally associated brand and then asked to answer the given question. A total of 220 questionnaires were distributed, of which 203 were collected and 187 were used for analysis, except for 16 questions that were not answered properly.

## 4.2. Constructs and Measurement Items

For the measurement of this study, operational definitions were made on each construct to meet the purpose of the study, and then measurement items were constructed based on previous studies. Questionnaires were constructed on a 5-point Likert scale. In addition to demographic variables, 32 items were designed for 10 constructs. The specific contents of the measurement items are shown in Table 2 below.

**Table 2.** Measurement items.

| Constructs | | Items | Researcher |
|---|---|---|---|
| Desire for belonging | Prestige sensitivity | I tend to consume the product in hopes that others will find out. I tend to consume the product to express my supremacy to others. I feel I'm ahead of others when I buy the product I want. I am willing to pay more money if I can buy the product I want. | Elliott [34] |
| | Connectedness | I tend to buy products to belong to the mainstream society. I tend to buy products to share the lifestyle of mainstream groups. I tend to buy products to share the values of mainstream groups. | Oberst et al. [28] |
| | Praise from others | I have been interested in the product in the desire to be recognized by people around me. I have experience purchasing products to match people around me. I have consumed the product in the hope that people around me will look positively towards me. | Abel et al. [15] |
| Anxiety of isolation | Being alienated | I feel like I'm being neglected by the group if I do not buy the products that people have. I would not be able to understand their values unless I buy the products they have. I would not be able to get along with them unless I buy the products they have. | Ankony and Kelley [55] |
| | Being ignored | I am afraid that if people do not buy the products they have, people will think of me as strange. I am afraid that when I do not buy a popular product, people around me will have a bad eye. I am likely to think of me as negative if I do not buy the products that people around me have. I am afraid that people will ignore me if I do not buy the products they have. | Lai et al. [25] |
| | Falling behind | I get the impression that I am outdated if I do not buy the products that people have. I get the impression that I have not been able to do it if I do not buy the products that people have. I am afraid that if I do not buy the products that people have, I will be out of trend. | Molden et al. [51] |
| Reinforcing stability | | After consuming this product, I seem to be more like a social mainstream lifestyle. After consuming this product, it seems that people around me are looking more positively about me. After consuming this product, I think that I am leading the trend myself. | Verhoef et al. [42] |
| Increasing concern | | I would have thought that if I did not buy this product, I would be lagging behind other people. If I did not buy this product, I would feel more alienated from the people around me. If I did not buy this product, I would have become more anxious about the people around me. | Watson [29] |
| Up-surging interest on culturally associated brand | | I have been curious about the culturally associated brand recently. I recently became interested in this culturally associated brand. I recently asked people around me about this culturally associated brand. | Liang and He [20] |
| Conformity consumption | | To make sure I buy the right brand, I often observe what others are buying and using. I frequently gather information from friends or family about a brand before I buy. When buying products, I generally purchase those brands that I think others will approve of. | Bearden et al. [4] |

## 5. Findings

### 5.1. Sample Characteristics and Correlations

This study divides the characteristics of 187 respondents who answered the questionnaire, as shown in Table 3. Of the respondents, male (98, 52.4%) was slightly higher than female (89, 47.6%). The respondents under 20 showed the highest rate of 66 persons (35.3%), followed by those in their 20s (52, 27.8%), 30s (40, 21.4%), and over 40s (29, 15.5%). The occupation of the respondents showed the highest proportion was private employees (63, 33.7%), followed by students (55, 29.4%). The demographic characteristics are detailed in Table 3 below.

The correlations between the constructs used in the study are shown in Table 4. In the correlation between the fear of missing out and the evaluation of culturally associated brands, the desire for belonging was found to have an overall significant relationship with the reinforcing stability. Connectedness was the highest with 0.40 ($p < 0.01$), followed by reinforcing stability. In the relationship between anxiety of isolation and increasing concern, being alienated showed the highest relevance, 0.36 ($p < 0.01$). On the other hand, the increasing concern (0.26, $p < 0.01$) was relatively higher than the reinforcing stability (0.20, $p < 0.01$) on the up-surging interest on culturally associated brands. In addition, up-surging interest on culturally associated brands was found to be highly correlated with conformity consumption (0.35, $p < 0.01$).

**Table 3.** Demographic characteristics (n = 187).

| Item | Characteristics | Frequency | Ratio |
|---|---|---|---|
| Gender | Male | 98 | 52.4% |
| | Female | 89 | 47.6% |
| Age | Under 20 | 66 | 35.3% |
| | 20–30 | 52 | 27.8% |
| | 30–40 | 40 | 21.4% |
| | Over 40 | 29 | 15.5% |
| Occupation | Students | 55 | 29.4% |
| | Private employees | 63 | 33.7% |
| | Public employees | 29 | 15.5% |
| | Homemakers | 15 | 8.0% |
| | Self-employed | 20 | 10.7% |
| | Other | 5 | 2.7% |

**Table 4.** Correlations among constructs.

| | ① | ② | ③ | ④ | ⑤ | ⑥ | ⑦ | ⑧ | ⑨ | ⑩ |
|---|---|---|---|---|---|---|---|---|---|---|
| ① | 1.00 | | | | | | | | | |
| ② | 0.42 ** | 1.00 | | | | | | | | |
| ③ | 0.36 ** | 0.41 ** | 1.00 | | | | | | | |
| ④ | 0.03 | 0.07 | 0.17 * | 1.00 | | | | | | |
| ⑤ | 0.12 | 0.13 | 0.19 ** | 0.51 ** | 1.00 | | | | | |
| ⑥ | 0.10 | 0.14 | 0.18 * | 0.23 ** | 0.26 ** | 1.00 | | | | |
| ⑦ | 0.33 ** | 0.40 ** | 0. 37 ** | −0.08 | 0.06 | −0.05 | 1.00 | | | |
| ⑧ | 0. 01 | 0.09 | 0.03 | 0.36 ** | 0.25 ** | 0.23 ** | −0.04 | 1.00 | | |
| ⑨ | 0.27 ** | 0.16 * | 0.01 | 0.20 ** | −0.03 | 0.11 | 0.20 ** | 0.26 ** | 1.00 | |
| ⑩ | 0.36 ** | 0.20 ** | 0.05 | 0.11 | −0.04 | −0.06 | 0.28 ** | 0.30 ** | 0.35 ** | 1.00 |
| Mean | 2.88 | 2.96 | 2.72 | 2.91 | 3.26 | 2.81 | 3.27 | 2.51 | 3.28 | 2.81 |
| S.D. | 0.83 | 0.87 | 0.87 | 0.94 | 0.80 | 0.88 | 0.91 | 0.97 | 1.07 | 1.01 |

① Prestige sensitivity, ② Connectedness, ③ Praise from others, ④ Being alienated, ⑤ Being ignored, ⑥ Falling behind, ⑦ Reinforcing stability, ⑧ Increasing concern, ⑨ Up-surging interest on culturally associated brand, ⑩ Conformity consumption, ** $p < 0.01$, * $p < 0.05$.

## 5.2. Measure Validation

In order to examine the reliability and validity of the measurement items used in the analysis, this study conducted confirmatory factor analysis (CFA). To this end, this study utilized the AMOS statistical package. As shown in Table 5, it was found that the convergent validity was secured by a *t*-value of at least 8.90 for each measurement item. In addition, the CR value is more than 0.70, and the AVE value, which means the variance extraction value, is more than 0.50, which can be regarded as a reliable category [56]. As a result of the analysis, it was confirmed that both the CR value and the AVE value presented in this study were satisfactory. On the other hand, in order to confirm the discriminant validity, this study examined the correlation coefficient between the square root of the AVE value and the variable. As shown in Table 5, the correlation coefficient was lower than the square root of the AVE value and the discriminant validity of the items used in this study was confirmed to be good.

**Table 5.** CFA results.

| Constructs | | Items | Loading | S.E | t-Value | Cronbach's $\alpha$ | CR | AVE |
|---|---|---|---|---|---|---|---|---|
| Desire for belonging | Prestige sensitivity | PS1 | 1.00 | - | - | 0.88 | 0.90 | 0.70 |
| | | PS2 | 1.06 | 0.08 | 12.65 | | | |
| | | PS3 | 0.94 | 0.08 | 11.65 | | | |
| | | PS4 | 0.94 | 0.08 | 11.71 | | | |
| | Connectedness | CN1 | 1.00 | - | - | 0.91 | 0.85 | 0.75 |
| | | CN2 | 0.99 | 0.06 | 15.51 | | | |
| | | CN3 | 0.99 | 0.06 | 15.88 | | | |
| | Praise from others | PO1 | 1.00 | - | - | 0.90 | 0.81 | 0.77 |
| | | PO2 | 0.89 | 0.06 | 15.08 | | | |
| | | PO3 | 0.94 | 0.07 | 14.35 | | | |
| Anxiety of isolation | Being alienated | BA1 | 1.00 | - | - | 0.88 | 0.88 | 0.68 |
| | | BA2 | 0.93 | 0.07 | 13.76 | | | |
| | | BA3 | 0.85 | 0.07 | 12.68 | | | |
| | Being ignored | BI1 | 1.00 | - | - | 0.85 | 0.78 | 0.80 |
| | | BI2 | 1.12 | 0.12 | 9.08 | | | |
| | | BI3 | 1.22 | 0.13 | 9.39 | | | |
| | | BI4 | 1.14 | 0.13 | 8.90 | | | |
| | Falling behind | FB1 | 1.00 | - | - | 0.88 | 0.89 | 0.81 |
| | | FB2 | 0.91 | 0.07 | 13.46 | | | |
| | | FB3 | 0.94 | 0.07 | 13.35 | | | |
| Reinforcing stability | | RS1 | 1.00 | - | - | 0.88 | 0.93 | 0.82 |
| | | RS2 | 0.96 | 0.07 | 13.47 | | | |
| | | RS3 | 0.97 | 0.07 | 13.31 | | | |
| Increasing concern | | IC1 | 1.00 | - | - | 0.91 | 0.92 | 0.74 |
| | | IC2 | 1.03 | 0.07 | 15.71 | | | |
| | | IC3 | 0.97 | 0.06 | 15.79 | | | |
| Up-surging interest on culturally associated brand | | UI1 | 1.00 | - | - | 0.91 | 0.88 | 0.69 |
| | | UI2 | 1.06 | 0.06 | 17.23 | | | |
| | | UI3 | 0.99 | 0.07 | 14.28 | | | |
| Conformity consumption | | CF1 | 1.00 | - | - | 0.86 | 0.87 | 0.75 |
| | | CF2 | 1.17 | 0.09 | 13.25 | | | |
| | | CF3 | 0.82 | 0.08 | 10.83 | | | |

Meanwhile, because the constructs used in this study were measured from the same respondents by the self-report response of the cross-sectional survey, Harman's one-factor test was performed to determine whether the common method bias could distort the interpretation of the results [57]. Harman's one-factor validation using exploratory factor analysis (EFA) was performed. A total of 10 factors with an Eigen value of 1 or more were derived from the EFA (principal components method) of

Varimax rotation, which were applied to 32 measurement items applied to the measurement model analysis. These factors accounted for 69.88% of the total variance, respectively. These results indicate that the results of this study are less likely to be distorted due to common method bias [57]. In this study, common method bias was confirmed by applying Harman's one-factor test through CFA. The results of the single factor model showed that 32 items were not loaded on a single factor and did not account for most of the covariance between the measured variables.

### 5.3. Empirical Results

This study examined the suitability of the structural equitation model before examining the causal relationship between concept variables. The model fit of the study was as follows: $\chi^2 = 677.10$ ($p < 0.01$), degree of freedom = 440, GFI = 0.84, AGFI = 0.80, NFI = 0.86, CFI = 0.94, IFI = 0.94, and RMSEA = 0.05. In general, if the value of $\chi^2$/degree of freedom is less than 3 and the GFI is greater than 0.90, AGFI is greater than 0.80, CFI is greater than 0.90, NFI is greater than 0.90, IFI is greater than 0.90, and RMSEA is between 0.05 and 0.08. [56]. In this study, although the GFI and NFI values are slightly lower, the other indicators are generally good, which is considered to be a relatively acceptable model (Table 6).

**Table 6.** Statistical summary of model fit.

| Model Goodness of Fit Indexes | Recommended Value | Results in This Study |
| --- | --- | --- |
| Chi-squared/degree of freedom | ≤3 | 1.53 |
| Goodness of Fit Index (GFI) | ≥0.90 | 0.84 |
| Adjusted Goodness of Fit Index (AGFI) | ≥0.80 | 0.80 |
| Normalized Fit Index (NFI) | ≥0.90 | 0.86 |
| Comparative Fit Index (CFI) | ≥0.90 | 0.94 |
| Incremental Fit Index (IFI) | ≥0.90 | 0.94 |
| Root Mean Squared Error of Approximation (RMSEA) | 0.05–0.08 | 0.05 |

The results of this study are shown in Figure 3. FoMO can be distinguished from two kinds of psychological traits, one is the desire for belonging and the other one is anxiety of isolation. The reinforcing stability caused from the desire for belonging and the increasing concern arise from the anxiety of isolation result in the up-surging of interest on culturally associated brands and, at last, lead to conformity consumption.

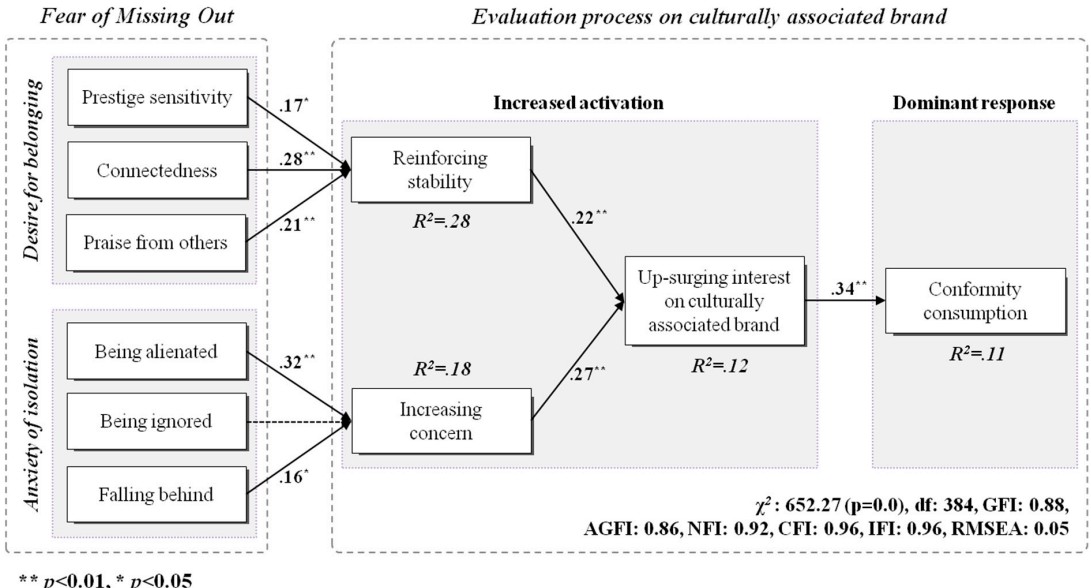

**Figure 3.** Causal effects among constructs.

Among the two kinds of paths that influence conformity consumption, the increasing concern that arises from anxiety of isolation has more significant influence on conformity consumption than the influence of reinforcing stability on conformity consumption through the desire to belong. Among the anxiety of isolation and increasing concern, being alienated had the highest coefficient value, indicating a significant relationship (β = 0.32, $p$ < 0.01). Being ignored was proven to have no statistically significant relationship with increasing concern and falling behind was found to have a significant impact (β = 0.16, $p$ < 0.05). Thus, Hypothesis 4 and Hypothesis 6 were supported, whereas Hypothesis 5 was rejected. In the relationship between the desire to belong and reinforcing stability, connectedness was found to have the highest coefficient (β = 0.27, $p$ < 0.01), followed by praise from others (β = 0.21, $p$ < 0.01) and prestige sensitivity (β = 0.17, $p$ < 0.05). As a result, Hypothesis 1, Hypothesis 2, and Hypothesis 3 were all supported.

The reason why connectedness and being alienated influenced the most reinforcing stability and increasing concern, respectively, is that consumers are most conscious of belonging to group. Many studies that have proved the relationship between the sense of belonging and stability as an important factor in determining stability [58,59]. The results of this study are also consistent with the findings of previous studies. Similarly, being alienated, which has the highest impact on increasing concern, can be interpreted in the same context as social belonging. According to social capital theory, people think of themselves as members of a large group and that they are connected and related to others, and are of greater value to the harmony of groups. This theory explains that interdependence among people has a direct impact on the prevention focused tendency in order to alleviate individual anxiety [60]. In other words, consumers can interpret it as a psychological feeling that is the main cause of perceived individual anxiety within the group is neglected.

On the other hand, reinforcing stability and increasing concern were found to be significantly related to up-surging interest on culturally associated brands and up-surging interest on culturally associated brands was found to have a strong influence on conformity consumption. Thus, Hypothesis 7, Hypothesis 8, and Hypothesis 9 were also supported. Among them, increasing concern showed a relatively higher coefficient value (β = 0.27, $p$ < 0.01) than reinforcing stability. In this process, the perceived up-surging interest on culturally associated brands has a strong influence on the conformity consumption (β = 0.34, $p$ < 0.01). The findings proved that consumers' level of interest in culturally associated brands ultimately plays a major role in bringing about conformity consumption.

## 6. Conclusions and Discussion

The implications of this study, based on the verification results, can be summarized as follows. First, unlike studies of conformity consumption that investigated factors from the aspects of brand characteristics and individual motivations, this study paid attention to a specific psychological trait, namely FoMO, and used it to explain an excessive conformity consumption phenomenon in the Korean market, which had been difficult to explain in previous studies. Moreover, the current study proposed an evaluation process for culturally associated brands based on a conceptual framework that showed that the increasing activation of stability and concern are strengthened through the presence of others following because of FoMO, and ultimately result in a conformity consumption response. The results indicate that consumers' attitude toward culturally associated brands emphasizes the desire or anxiety of individuals to participate in the mainstream group. In other words, the willingness to belong to the mainstream group or the fear of being excluded from the mainstream group played a crucial role in the consumption of culturally associated brands.

Second, this study distinguished between two kinds of psychological traits of FoMO, based on previous studies. One is the desire for belonging, which refers to the one who experiences FoMO as having a strong need for interpersonal attachments, such as the need to identify with their status and need for recognition from others, as a fundamental human motivation. The other one is anxiety of isolation, which describes the fear of being broken away and isolated from the mainstream group.

According to these psychological traits, we divided FoMO into six dimensions, by which it is possible to carry out more accurate and more significant measurements than previous studies.

Third, the results of this study showed that being alienated was a major cause of conformity consumption of expensive clothing among Korean consumers. The fear of being alienated from the mainstream group results in a high level of increasing concern regarding brands and, through the emotional change, consumers tend to have more interest in culturally associated brands, which leads to conformity consumption. The finding is meaningful in that it investigates the factors leading to the excessive conformity consumption phenomenon in the Korean market, which had been insufficiently explained using previous methods.

According to the findings of the current research, we present the following managerial implications. First, since FoMO has a significant positive effect on conformity consumption of culturally associated brands, especially leading to the collective following-up conformity tendency on certain groups, it is necessary to provide products and services for specific consumer groups following the cultural trends. For example, ZIGZAG in Korean and Xiaohongshu in China provide services to users with information on what content is most important to users and what kind of real-time queries are classified by gender or age. In addition, Xiaohongshu is highly acclaimed for customizing products and services that are of interest to particular groups and then advertising them to consumers. On the other hand, it is strategic to provide the advertising message that something is desirable for consumption to consumers who are in a high level of FoMO, especially to the consumers of young groups.

Second, due to the culturally associated brands not only providing features of general products, but also reflecting specific cultural values, in addition to the characteristics of products or services, companies also need to pay attention to the phenomenon of cultural consumption. The effect of cultural consumption should be emphasized to cater to the needs of consumers and stimulate potential consumption. In particular, information dissemination is speeding up presently and, as a result, it is necessary for managers to be aware of various effects, such as cultural influences, when conducting marketing activities.

## 7. Limitations and Future Research

Despite these implications, this study has several limitations. First, a comparative study of product categories should be conducted to evaluate culturally associated brands. This study selected expensive clothing brands which are easy to be exposed to the outside in order to investigate the conformity consumption of consumers. However, there is a possibility that conformity consumption, as well as the FoMO phenomenon, may be different depending on product categories, such as public products, private products, luxury goods, and necessities [55]. In future studies, it is possible to provide more concrete implications if the comparative studies are carried out that reflect the characteristics of these products.

Second, attitudes to culturally associated brands are likely to change along with different characteristics among consumers. There will be a difference depending on the consumption propensity of the individual, such as the different levels of involvement with a certain brand, consumption tendency, inherent cultural value, and lifestyle [61]. If future research includes comparative studies that reflect these consumer characteristics, it would be able to more specifically grasp the FoMO phenomenon of the mainstream group.

Third, our survey primarily captures a cross-sectional view of constructs. It is possible that Korean consumers' perceptions of the tested culturally associated brands would be different after a certain period. Thus, a longitudinal study is necessary to investigate time-variant impacts on the assessments of culturally associated brands. Although these limitations exist, it is expected that this study makes an academic contribution to the evaluation of culturally associated brands and is a useful study for practitioners as well.

**Author Contributions:** I.K., H.C., and J.S. contributed significantly to the conception of the study, data collection, analysis, and to preparing the manuscript.

**Funding:** This research was funded by [Kyung Hee University] grant number [KHU-20181287].

**Conflicts of Interest:** The authors declare no conflict of interest.

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
