# Peer review of "Conformity Consumption Behavior and FoMO"

_sustainability, doi:10.3390/su11174734_

Round 1
Reviewer 1 Report
This study has merit in that it tackles an interesting question and attempts to uncover new details about the relationship of FoMO to conformity consumption which should be of interest to a wide audience of readers.
Below are several suggestions for improvement:
The paper needs to be revised by a professional copy editor.
In my opinion, the conceptualization of FoMO as presented is problematic. For one, several of the dimensions appear to capture opposing endpoints of a single dimension rather than two distinct dimensions (such as praise from others and disapproval from others). More significantly, the overall approach to the way in which FoMO is conceptualized seems problematic. More specifically, the items/dimensions appear to capture other traits and variables that explain consumer behavior and are likely to influence the amount FoMO one experiences but do not represent FoMO themselves. For example, prestige sensitivity does not represent how much you feel you are missing out. It represents how much you care about prestige which in turn could lead one to experience a fear of missing out more.
it is interesting to see how you have included an approach motivation (stability) and an avoidance motivation (concern) in your model. However, it still seems you have not actually captured FoMO in any of your measures.
It would be interesting to see how collecting the data again with a FoMO scale such as the 10-item one used by Przybylski et al (2013) would improve the contribution of this paper and our understanding of the relationships between variables identified in the current manuscript.
Author Response
As the reviewer pointed out, the overall correction was made to solve the grammatical problems in the text. The revised details are shown in the attached file below.

Reviewer 2 Report
The reviewed paper is of scientific nature. It is based on literature studies and results of empirical research. The subject area discussed in the paper is important and topical. The structure of the paper is clear.
I would suggest to develop:
- the conclusions and discussion,
- the description of research limitations and future research.
Author Response
The revised details are shown in the attached file below.

Round 2
Reviewer 1 Report
Thanks for your response. Unfortunately, the logic justifying the conceptualization of FoMO as presented in the study still seems flawed.
As an additional example, connectedness is presented as a dimension of FoMO while the literature suggests that the extent to which someone is “connected” can lead to FoMO. As presented, the dimensions of FoMO in this paper appear to be more like antecedents/correlates to FoMO.
Since it does not really appear the authors have captured FoMO (but instead antecedents of it), perhaps the model could be re-labeled so that FoMO (as a construct) is simply removed. The relationships represented in the model seem founded, just mislabeled.
Author Response
2nd round comments
Thank you for your review. We revised the article to reflect your comments. The revised details are as follows.
Comment: As an additional example, connectedness is presented as a dimension of FoMO while the literature suggests that the extent to which someone is “connected” can lead to FoMO. As presented, the dimensions of FoMO in this paper appear to be more like antecedents/correlates to FoMO.
Response: As the reviewers pointed out, we determined that the explanation for the connectedness presented in the FoMO dimension could be ambiguous. Therefore, we reviewed and revised the previous studies that dealt with FoMO, and based on this, we clearly explained what dimensions the FoMO is divided into than the previous version (p.3~4). In addition, as the reviewer commented, we also thought that the connectedness presented in the text could be regarded as an antecedent of FoMO. To complement this, we have explained in more detail why connectedness is meaningful as a sub-dimension of FoMO (p.4~5). Furthermore, we have added specific explanations for what not only connectedness but other factors also mean as a sub-dimension of FoMO (e.g. prestige sensitivity).
Comment: Since it does not really appear the authors have captured FoMO (but instead antecedents of it), perhaps the model could be re-labeled so that FoMO (as a construct) is simply removed. The relationships represented in the model seem founded, just mislabeled.
Response: One of the main purposes of this study is to extract the sub-dimensions of FoMO and to analyze how these sub-dimensions affect consumer attitude changes. However, as the reviewer pointed out, we decided that we needed to complement the logical development. To this end, we explained in more detail how the evaluation process reflects the FoMO phenomenon (p. 6; Evaluation process of culturally associated brand based on FoMO). Through this, the causal relationship between FoMO and the change in consumer attitude is more clearly organized.
In addition, we revised the text to make the logical development more clear.
Round 3
Reviewer 1 Report
Thanks to the authors. In my opinion, the manuscript now much more clearly justifies the basis for the conceptualization of the model/operationalization of the FoMOconstruct and makes an interesting contribution to the literature. One direction for future research might be to incorporate alternative operationalizations/measures of FoMO (suggested in other disciplines) to test the sensitivity of the current results to the method.
This manuscript is a resubmission of an earlier submission. The following is a list of the peer review reports and author responses from that submission.
Round 1
Reviewer 1 Report
p.p1 {margin: 0.0px 0.0px 0.0px 0.0px; font: 12.0px 'Helvetica Neue'; color: #454545} p.p2 {margin: 0.0px 0.0px 0.0px 0.0px; font: 12.0px 'Helvetica Neue'; color: #454545; min-height: 14.0px}Thank you for giving me the possibility of reviewing this paper. I hope the authors find my comments productive and that it will help them to improve their research work.
The objective of this study is to investigate the FOMO in offline environments as a result of cultural consumption of brands.
The objective is clearly stated.
However it is not clear why it is necessary to implement the FOMO in offline environments. There are sufficient methods to research offline. Eventough the application of FOMO offline is a novelty, why doing so? have other authors done it before?
Also behavior online and offline differs so it is important to choose the right method.
The authors need to justify their decision about the method based on previous literature.
Reviewer 2 Report
In the assessment of the paper submitted for the review, I specifically focused on the discussed issues, applied research procedure, substantive content of the paper and its structure.
The topic of this paper seems to be interesting.
The considerations conducted in the paper are focused on such categories as: fear of missing out, culturally associated brand, conformity consumption, evaluation process on cultural associated brand.
The subject area discussed in the paper should be considered important. The value of the paper results from appropriate combination of literature studies with the results of an empirical study. The structure of the paper is clear.
It is specifically recommended to:
- clarify the purpose of the paper (in the abstract of paper),
-specify the managerial implications.
Reviewer 3 Report
The introduction is predominantly focused on FoMo but it does not really make it clear what it is in an offline situation. It states that FoMo is a personality trait which it isn’t really. It needs to be clearer in defining what it is, it is something individuals feel due to their what purchase/lack of purchase and their group’s reaction to them as a result of their purchase?
Table 1 is not especially helpful. What is the reader supposed to extract from this table? Are the authors arranged in a random order or an order according to similarity or something else? It needs to make it clear what research has identified as behaviours resulting from FoMo in an online conext and then say what FoMo is off line and what behaviours it might lead to. It needs to be more in depth and more structured within the sections. It needs to be clearer in the discussion and evaluation of the dimensions as some of these are selected and some are rejected for the study and it needs to be made clear why.
It discusses cultural associated brands but it does not really make it clear what makes a cultural associate brand different from a brand.
Section 4 is a bit confused, it mentions attitudes, positive and negative emotions and stability and concern but there is a lack of connection between the terms. The terms need to be more fully explained. What exactly is being looked at and measured? Figure 1 needs a more precise explanation. It has lines above the boxes which are in need of explanation.
It needs to be careful regarding explanations of the terms used and their measurement. Conformity is defined as compliance with group norms but this is not what is actually measured. The measures do not mention cultural associated brands.
The vagueness of definitions of terms and the similarity of terms makes this paper makes it a bit problematic.
Methodology
The choice of cultural associated brands needs better justification. The detail of the questionnaire needs to be fuller, so what was the phrasing of the questions in relation to the products, what additional questions were asked? Why were those particular scales used? In terms of how the sample was obtained where was it posted or how was it distributed to obtain the sample?
I think there is a bit of an issue with the terms used in the literature and the measures used. The up-surging interest on cultural associated brands measures are not readily apparent form the reference provided. Is the measure for conformity consumption actually measuring conformity consumption?
Findings
A bit more detail could be provided on the factor analysis in terms of what was done. There’s a bit of a mix of results with the goodness of fit figures which could be discussed a bit more in terms of why some were used and not others.
Discussion
The discussion is ok. It states that FoMo is a manageable resource which is a bit vague. It is not even clear what FoMo in an offline situation is.